# Adverse events associated with the delivery of telerehabilitation: A scoping review protocol

**Thomas Yau**[1], **McKyla McIntyre**[2,3], **Josh Chan**[4], **Damanveer Bhogal**[1],
**Angie Andreoli**[2], **Mark Bayley**[1,2,5,6,7], **Carl Froilan D. Leochico**[1,8,9], **Ailene Kua**[2,5],
**Meiqi Guo**[2,3‡], **Sarah Munce**[2,5,6,7,10‡*]

**1** Temerty Faculty of Medicine, University of Toronto, Toronto, ON, Canada, **2** Toronto Rehabilitation Institute-University Health Network, Toronto, ON, Canada, **3** Faculty of Medicine, Department of Medicine, Division of Physical Medicine and Rehabilitation, University of Toronto, Toronto, ON, Canada, **4** Western University, London, ON, Canada, **5** KITE Research Institute, Toronto Rehabilitation Institute-University Health Network, Toronto, ON, Canada, **6** Department of Occupational Science and Occupational Therapy, University of Toronto Rehabilitation Sciences Institute, University of Toronto, Toronto, ON, Canada, **7** Institute of Health Policy, Management and Evaluation, University of Toronto, University of Toronto, Toronto, ON, Canada, **8** St. Luke's Medical Center, Global City, Philippines, **9** Philippine General Hospital, University of the Philippines Manila, Manila, Philippines, **10** Rehabilitation Sciences Institute, University of Toronto, Toronto, ON, Canada

☯ These authors contributed equally to this work.
‡ MG and SM contributed equally as Co-senior authors.
* sarah.munce@uhn.ca

**Data Availability Statement:** No datasets were generated or analysed during the current study. All relevant data from this study will be made available upon study completion.

## Abstract

### Objective

This scoping review aims to map the existing research on adverse events during the delivery of telerehabilitation.

## Introduction

Telerehabilitation, a subset of telemedicine, has gained traction during the COVID-19 pandemic as a means to deliver rehabilitation services remotely. However, there exists a research gap as there has yet to be any scoping review, systematic review, or meta-analysis published to identify and summarize the current primary research on adverse events related to telerehabilitation as a whole. It is important to understand how adverse events, such as falls during physiotherapy or aspiration pneumonia during speech language pathology sessions, are associated with telerehabilitation delivery. This will help to identify key limitations for optimizing telerehabilitation delivery by allowing for the development of key risk-mitigation measures and quality indicators. It can also help improve the uptake of telerehabilitation among clinicians and patients. This review aims to fill this research gap by conducting a search of published literature on adverse events in telerehabilitation. Anticipated key findings of this scoping review include identifying the characteristics and frequencies of adverse events during telerehabilitation, the patient populations and types of telerehabilitation associated with the most adverse events, and the quality of reporting of adverse events.

**Funding:** This work was supported by the University Health Network Foundation (TY). The funder did not and will not have a role in study design, data collection and analysis, decision to publish, or preparation of the manuscript.

**Competing interests:** The authors have declared that no competing interests exist.

## Methods

The review follows the Joanna Briggs Institute (JBI) methodological framework and adheres to the Preferred Reporting Items for Systematic Reviews and Meta-Analyses Extension for Scoping Reviews (PRISMA-ScR) guidelines. The review protocol has been registered and published on Open Science Framework. A comprehensive search strategy was implemented across multiple databases (MEDLINE ALL, EMBASE, APA PsycINFO, CENTRAL, and CINAHL). All stages (screening, extraction, and synthesis) will be conducted in duplicate and independently, with data extraction following the TIDieR framework, along with authors, year of publication (before or after COVID), population and sample size, and specific mode/s of telerehabilitation delivery. For synthesis, data will be summarized quantitatively using numerical counts and qualitatively via content analysis. The data will be grouped by intervention type and by type of adverse event.

## Inclusion Criteria

This scoping review will include qualitative and quantitative studies published between 2013 and 2023, written in English, and conducted in any geographic area. All modes of telerehabilitation delivery (asynchronous, synchronous, or hybrid) will be included. Systematic reviews, meta-analyses, commentaries, protocols, opinion pieces, conference abstracts, and case series with fewer than five participants will be excluded.

## Introduction

Telerehabilitation is a subset of telemedicine connecting rehabilitation providers and patients over a distance [1]. The use of telerehabilitation rapidly increased during the COVID-19 pandemic to deliver rehabilitation services while preventing disease transmission [2]. Telerehabilitation can provide services to those who would not normally be able to access traditional rehabilitation, such as those living in remote communities or patients with disabilities, which hinder participation in in-person sessions [3]. The convenience of telerehabilitation may also lead to higher attendance rates for individuals with busy schedules or other commitments. It may also be a less expensive alternative to in-person rehabilitation due to decreased travel expenses [4]. Multiple systematic reviews have shown the effectiveness of telerehabilitation; for instance, Dias et al. found high-quality evidence that telerehabilitation was not different from other interventions for adults with physical disabilities in terms of improvements in pain, physical function, and long-term quality of life [5–9]. However, questions remain about potential limitations of telerehabilitation, particularly regarding its safety compared to traditional in-person rehabilitation. Due to the remote nature of telerehabilitation, patients cannot receive immediate physical assistance from rehabilitation providers if they experience an adverse event. Adverse events are defined as "negative consequences of care that result in unintended injury or illness which may or may not have been preventable" [10]. For instance, they may include falls during physiotherapy or aspiration pneumonia due to speech language pathology swallowing assessments [11,12]. There can also be safety risks related to privacy, as personal health information is being transmitted across digital platforms. There is a paucity of research surrounding the patient safety of telerehabilitation, potentially contributing to its limited uptake among clinicians and patients [13]. While many individual studies include safety data,

there exists a research gap as there has yet to be any synthesis of the existing literature that summarize the currently available research on adverse events related to telerehabilitation. There has been a prior scoping review on measures to ensure safety during telerehabilitation for patients with stroke, specifically, but the current review differs as it focuses on adverse events and encompasses all health/chronic conditions that could be served by telerehabilitation [14]. This scoping review aims to conduct a systematic search of published literature on adverse events during the delivery of telerehabilitation and map out the extent of existing research. The WHO recognizes patient safety as a global health priority, and notes that investing in patient safety is important for health outcomes, cost reduction related to patient harm, and health system efficiency [15]. It is important to understand how adverse events are associated with telerehabilitation delivery, so that safety precautions and risk-mitigation measures can be thoughtfully planned and implemented, to optimize its uptake and delivery. Knowledge of the safety of telerehabilitation can help patients make more informed decisions, aid in clinical and funder decision-making, and inform safety quality indicators for telerehabilitation.

### Review question

What is the extent of literature on adverse events associated with the delivery of telerehabilitation?

## Methods

This protocol will adhere to the Joanna Briggs Institute (JBI) methodological framework for scoping reviews, which provides guidance on the outline of the review, inclusion criteria (i.e. PCC), search strategy, extraction, presenting and summarizing the results, and any potential implications of the findings for research and practice [16]. The reporting of the scoping review will adhere to the Preferred Reporting Items for Systematic Reviews and Meta-Analyses Extension for Scoping Reviews (PRISMA-ScR) guidelines, to ensure all the components of a high-quality scoping review are completed [17] and a filled checklist will be viewable in S1 Appendix. Our team includes members with extensive experience in scoping reviews and telerehabilitation.

### Protocol and registration

The protocol was registered and published on Open Science Framework on June 26, 2023 (Registration DOI: https://doi.org/10.17605/OSF.IO/C3ZHQ).

### Eligibility criteria

Various study designs will be considered in this scoping review (e.g., experimental, quasi-experimental, observational, qualitative, mixed and multiple methods). However, systematic reviews, meta-analyses, commentaries, protocols, opinion pieces (editorials), abstracts from conferences, and case series of <5 participants will not be included. Studies will be limited to those published between 2013–2023, because a study by Zheng et al. found that 2013 was the start of a more significant development period of telerehabilitation, with only a few papers on telerehabilitation published prior [18]. Additionally, the year 2013 marked the emergence of video communication technologies such as Zoom and Google Hangout that are commonly used in telerehabilitation today, which will ensure that the review's results are relevant to the current practice of telerehabilitation [19]. Studies must be written in the English language but can be from any geographic area. All modes of delivery for telerehabilitation (asynchronous, synchronous, or hybrid) are eligible.

## Search strategy

Search strategies were developed by a librarian with experience searching the health sciences literature and conducting systematic and scoping reviews. The following databases were searched on the Ovid platform: MEDLINE ALL, EMBASE, APA PsycINFO, and Cochrane Central Register of Controlled Trials (CENTRAL). The Cumulative Index to Nursing and Allied Health Literature (CINAHL) database was searched on the EBSCOHost platform. An initial strategy was created in MEDLINE ALL and sent to the team for review. Once the test strategy for MEDLINE ALL was agreed upon, the librarian sought out a volunteer librarian to provide a Peer Review of Electronic Search Strategies (PRESS) review.

The MEDLINE ALL search strategies were translated using the command language, controlled vocabulary, and appropriate search fields for each database and search platform. Search terms included Medical Subject Headings (MeSH), EMTREE terms, American Psychological Association thesaurus terms, and CINAHL headings and text words to capture concepts and synonyms of telerehabilitation and adverse events. Results were limited to the English language and the publication period from 2013 to present. The full MEDLINE ALL search strategy can be viewed in S2 Appendix.

## Study/Source of evidence selection

All identified citations will be imported into EndNote to remove duplicates. They will then be transferred into Covidence. All rounds of screening will be completed in duplicate and independently. After completing a pilot test, titles and abstracts will be screened. Sources that meet inclusion criteria will be retrieved in full. This will then be followed by a round of screening based on full texts. The results of the search and study inclusion process will be illustrated in a PRISMA-ScR flow diagram.

## Data extraction

Using Covidence, data will be extracted from the papers in duplicate and independently. The data extracted will follow the Template for Intervention Description and Replication (TIDieR) framework, including name of intervention, rationale, materials used, procedures, provider, mode of delivery, location, period of time, tailoring, modification over the course of study, and adherence [20]. It also will include specific details including authors, year of publication (before or after COVID), population and sample size, specific mode/s of telerehabilitation delivery (synchronous: videocall, phone call, instant messaging, web-based such as using either virtual reality or augmented reality; asynchronous: text/ audio/ video messaging, e-mails, on-demand resources; hybrid: combination of any synchronous and asynchronous methods), and outcome measures (social/psychological/physical adverse events, including severity if available). The extraction form can be viewed in S3 Appendix. The data extraction form will be pilot tested in duplicate and independently. Quality/risk of bias assessment will not be completed as this is not the purpose of a scoping review.

## Data analysis and presentation

Data from this scoping review will be summarized quantitatively using numerical counts and qualitatively via content analysis, based on best practices for reporting of scoping reviews [21]. The data will be grouped by intervention type and by type of adverse event (physical, social, psychological), and coded and analyzed manually. Numerical counts and content analysis will be used to reveal trends in the data such as the most common method of telerehabilitation, the

health condition with the most adverse events, and the frequency of different types of adverse events. Synthesis will occur in duplicate and independently.

## Discussion

The dissemination plan of this review includes traditional knowledge translation approaches of journal publication and conference presentations. This scoping review will be published in a relevant peer-reviewed academic journal such as PLOS One, Annals of Physical and Rehabilitation Medicine, Journal of Telemedicine and Telecare, or Telemedicine and eHealth, as they are reputable journals that publish articles within the same field as our review. This would help reach healthcare professionals, researchers, policymakers, and other relevant knowledge users who are actively searching for papers related to telerehabilitation and patient safety. The results from this review will be presented at conferences such as the American Telemedicine Association (ATA) Annual Conference, International Society of Physical and Rehabilitation Medicine (ISPRM) World Congress, American Congress of Rehabilitation Medicine, or International Conference on Telemedicine and Telehealth. This will further help reach the intended audience, including those who are not actively searching for papers on the topic. We will also leverage social media platforms to increase the visibility of the research article and reach a broad population who may not necessarily be up to date with academic journals or related conferences. This paper will also be presented at the Canadian Telerehabilitation Community of Practice, which is a forum for frontline clinicians. We will also engage with patient partners to elucidate the most effective ways of dissemination to this important knowledge user group.

This review has a number of strengths. For instance, all study phases (screening, extraction, synthesis) will be completed in duplicate and independently. This review is also guided by the JBI methodological framework for scoping reviews and will adhere to the PRISMA-ScR reporting guidelines [16,17]. These frameworks and guidelines were developed by internationally recognized experts in scoping review methodology for the development of high-quality scoping reviews. Following these recommendations will ensure all the components of a rigorous scoping review are completed. Furthermore, our team includes members with extensive experience in telerehabilitation and conducting scoping reviews.

Expected limitations of the study design are the exclusion of gray literature and studies not published in the English language. Excluding studies not published in the English language may reduce our understanding of telerehabilitation-related adverse events in limited resource countries. The exclusion of gray literature may lead to purely clinical settings and very recent data to be excluded from the review. In addition, no risk of bias assessment will be conducted as the aim of a scoping review is to map out the extent of literature. This means that this review will provide an overview of the literature and identify knowledge gaps, but will not assess the methodological quality or risk of bias of the individual studies, as that is not the purpose of a scoping review [17].

This scoping review will provide useful insights at an important juncture in time—there has been a surge in popularity of telerehabilitation due to the COVID-19 pandemic, and now patients, providers, funders and governments are contemplating how rehabilitation should be delivered in the future. Understanding adverse events related to telerehabilitation will help to identify key limitations for optimization, allowing for the development of necessary risk-mitigation measures and quality indicators. A greater understanding of the safety and optimization of telerehabilitation can help influence government funding and guide policymakers regarding whether telerehabilitation should be offered as part of standard care, which is important as a lack of leadership and organizational support has been found to hinder implementation of telerehabilitation [22]. These findings may also inform resource allocation decisions such that

safer forms of telerehabilitation can be prioritized for funding. The results may also suggest the need for telerehabilitation competency training for healthcare workers to ensure quality and safe care. The scoping review results will also inform researchers on how to optimize the safety of existing and emerging telerehabilitation technologies. From a patient perspective, findings from this review may aid in decision-making regarding participation in telerehabilitation.

## Supporting information

**S1 Appendix. PRISMA-P-checklist filled.**
(DOC)

**S2 Appendix. MEDLINE(R) ALL 1946 to June 22, 2023 search strategy.**
(DOCX)

**S3 Appendix. Extraction form.**
(DOCX)

## Acknowledgments

We would like to thank Thomasin Adams-Webber, Healthsearch, Library and Information Services, University Health Network, Toronto, Ontario, Canada, for the development of the search strategies, and Emilia Main, Healthsearch, Library and Information Services, University Health Network, Toronto, Ontario, Canada for the PRESS review.

## Author Contributions

**Conceptualization:** McKyla McIntyre, Angie Andreoli, Mark Bayley, Carl Froilan D. Leochico, Ailene Kua, Meiqi Guo, Sarah Munce.

**Funding acquisition:** McKyla McIntyre.

**Methodology:** Thomas Yau, McKyla McIntyre, Josh Chan, Damanveer Bhogal, Angie Andreoli, Mark Bayley, Carl Froilan D. Leochico, Ailene Kua, Meiqi Guo, Sarah Munce.

**Project administration:** Ailene Kua, Meiqi Guo.

**Supervision:** Meiqi Guo, Sarah Munce.

**Writing – original draft:** Thomas Yau, Josh Chan.

**Writing – review & editing:** McKyla McIntyre, Damanveer Bhogal, Angie Andreoli, Mark Bayley, Carl Froilan D. Leochico, Ailene Kua, Meiqi Guo, Sarah Munce.

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
