## [Decision Letter · Decision Letter 0]

22 Oct 2023

PONE-D-23-22135Adverse events associated with the delivery of telerehabilitation: A scoping review protocolPLOS ONE

Dear Dr. Munce,

Thank you for submitting your manuscript to PLOS ONE. After careful consideration, we feel that it has merit but does not fully meet PLOS ONE’s publication criteria as it currently stands. Therefore, we invite you to submit a revised version of the manuscript that addresses the points raised during the review process.

We look forward to receiving your revised manuscript.

Kind regards,

Kalyana Chakravarthy Bairapareddy, PhD

Academic Editor

PLOS ONE

Journal Requirements:

Additional Editor Comments:

Authors are requested to resubmit the manuscript after addressing the comments/ suggestions given by the reviewers.

Reviewers' comments:

Reviewer's Responses to Questions

**Comments to the Author**

1. Does the manuscript provide a valid rationale for the proposed study, with clearly identified and justified research questions?

Reviewer #1: No

Reviewer #2: Partly

2. Is the protocol technically sound and planned in a manner that will lead to a meaningful outcome and allow testing the stated hypotheses?

Reviewer #1: No

Reviewer #2: Yes

3. Is the methodology feasible and described in sufficient detail to allow the work to be replicable?

Reviewer #1: No

Reviewer #2: Yes

4. Have the authors described where all data underlying the findings will be made available when the study is complete?

Reviewer #1: Yes

Reviewer #2: No

5. Is the manuscript presented in an intelligible fashion and written in standard English?

Reviewer #1: Yes

Reviewer #2: Yes

6. Review Comments to the Author

You may also provide optional suggestions and comments to authors that they might find helpful in planning their study.

Reviewer #1: Reviewer's Report

Title: Adverse events associated with the delivery of telerehabilitation: A scoping review protocol

Abstract:

1. The abstract lacks a clear statement of the research gap and the significance of addressing adverse events in telerehabilitation. It should explicitly mention why understanding adverse events in telerehabilitation is important.

2. The abstract does not provide a concise summary of the key findings or anticipated outcomes of the scoping review. Including this information would help readers quickly grasp the scope and purpose of the study.

Introduction:

1. The introduction provides a general overview of telerehabilitation but lacks a strong narrative that directly leads to the research question. It should better establish the need for investigating adverse events in telerehabilitation.

2. While the introduction mentions the benefits of telerehabilitation, it could benefit from acknowledging some potential drawbacks or limitations, which would add context to the research question.

3. The citation of specific systematic reviews and their findings on the effectiveness of telerehabilitation is somewhat vague. It would be more informative to provide key findings or statistics from these reviews to support the assertion that telerehabilitation is effective.

4. The introduction could be more concise and focused on setting up the research question and rationale.

Methods:

1. While the methods section mentions adherence to the JBI framework and PRISMA-ScR guidelines, it lacks a detailed explanation of how these frameworks will be applied in the review. Clarification on the specific steps and criteria used would enhance the transparency of the review process.

2. The inclusion criteria mention studies published between 2013 and 2023, but the rationale for this time frame is not clearly explained. Providing a justification for this specific time frame would improve the clarity of the methodology.

3. The search strategy is briefly mentioned but lacks the actual search terms or keywords used. Including the search terms would make the methodology more reproducible.

4. The discussion of data analysis and presentation is limited. It would be helpful to provide more information on how the data will be synthesized quantitatively and how content analysis will be conducted.

5. The mention of contacting authors for missing data raises questions about how this process will be carried out and how it might affect the review's completeness. A brief explanation of the approach to contacting authors would be beneficial.

Discussion Section:

1. While the dissemination plan is mentioned, it lacks specificity regarding how the results will be translated to different audiences (healthcare professionals, researchers, policymakers, etc.). It would be beneficial to outline concrete strategies or channels for knowledge translation to ensure that the research reaches its intended audience effectively.

2. The manuscript lists several potential journals and conferences for publication and presentation, but it would be more informative to explain the rationale behind the selection of these specific outlets. Why were these journals and conferences chosen, and how do they align with the scope and goals of the review?

3. The discussion section mentions the strengths of the review, such as the use of the JBI framework, adherence to reporting guidelines, and the expertise of the review team. However, it would be helpful to expand on how these strengths contribute to the rigor and quality of the review. How will the JBI framework enhance the review process, for example?

4. While the expected limitations are briefly mentioned, the discussion could benefit from further exploration of how these limitations might impact the review's findings and implications. For instance, how might the exclusion of gray literature affect the comprehensiveness of the review?

5. The discussion mentions that no risk of bias assessment will be conducted due to the scoping review's nature. While this is acceptable for a scoping review, it would be helpful to briefly explain why assessing risk of bias is not applicable in this context.

6. The discussion outlines the importance of understanding adverse events related to telerehabilitation, especially in the context of the COVID-19 pandemic. However, it could be strengthened by providing specific examples of how these findings might inform government funding or guide policymakers. Concrete examples would enhance the practical implications of the review.

Overall, the discussion section could be improved by providing more specific details regarding knowledge translation, rationale for journal and conference selection, and a deeper exploration of the implications of expected limitations. Additionally, offering concrete examples of how the review's findings could impact decision-making and policy would enhance the section's clarity and impact.

Reviewer #2: The research theme is good and has high clinical significance. The protocol follows the standard guidelines. It would be interesting to see the outcomes of the proposed scoping review.

7. PLOS authors have the option to publish the peer review history of their article (what does this mean?). If published, this will include your full peer review and any attached files.

Reviewer #1: **Yes: **RAVI SHANKAR YERRAGONDA REDDY

Reviewer #2: No

---

## [Author Response · Author response to Decision Letter 0]

19 Dec 2023

December 19, 2023

Subject: Revision submission of our manuscript, “Adverse events associated with the delivery of telerehabilitation: A scoping review protocol” for consideration for publication in PLOS ONE.

Dear Reviewers:

Please find enclosed the revised manuscript, “Adverse events associated with the delivery of telerehabilitation: A scoping review protocol” for consideration for publication as a Study Protocol in PLOS ONE. We would like to thank you for the opportunity to resubmit a revised copy of this manuscript. We would also like to take this opportunity to express our thanks to the reviewers for the positive feedback and helpful comments for correction or modification. The manuscript has been revised to address the reviewers’ comments which are marked using track changes. We also provided our response to their comments below. The revisions have been developed in consultation with all coauthors, and each author has given approval to the final form of this revision. 

Reviewer's Responses to Questions

Comments to the Author

Abstract:

Comment 1: The abstract lacks a clear statement of the research gap and the significance of addressing adverse events in telerehabilitation. It should explicitly mention why understanding adverse events in telerehabilitation is important.

Response 1: Thank you for pointing this out. We have edited the abstract to clearly express this:

Telerehabilitation, a subset of telemedicine, has gained traction during the COVID-19 pandemic as a means to deliver rehabilitation services remotely. However, there exists a research gap as there has yet to be any scoping review, systematic review, or meta-analysis published to identify and summarize the current primary research on adverse events related to telerehabilitation as a whole. It is important to understand how adverse events, such as falls during physiotherapy or aspiration pneumonia during speech language pathology sessions, are associated with telerehabilitation delivery. This will help to identify key limitations for optimizing telerehabilitation delivery by allowing for the development of key risk-mitigation measures and quality indicators. It can also help improve the uptake of telerehabilitation among clinicians and patients. This review aims to fill this research gap by conducting a search of published literature on adverse events in telerehabilitation.

Comment 2: The abstract does not provide a concise summary of the key findings or anticipated outcomes of the scoping review. Including this information would help readers quickly grasp the scope and purpose of the study.

Response 2: Thank you for the recommendation. We have added the following sentences to the abstract:

Anticipated key findings of this scoping review include identifying the characteristics and frequencies of adverse events during telerehabilitation, the patient populations and types of telerehabilitation associated with the most adverse events, and the quality of reporting of adverse events.

Introduction:

Comment 1: The introduction provides a general overview of telerehabilitation but lacks a strong narrative that directly leads to the research question. It should better establish the need for investigating adverse events in telerehabilitation.

Response 1: Thank you for this feedback; we agree that a strong narrative leading to the research question would improve the introduction section. We have edited it as follows:

Telerehabilitation is a subset of telemedicine connecting rehabilitation providers and patients over a distance [1]. The use of telerehabilitation rapidly increased during the COVID-19 pandemic to deliver rehabilitation services while preventing disease transmission [2]. Telerehabilitation can provide services to those who would not normally be able to access traditional rehabilitation, such as those living in remote communities or patients with disabilities, which hinder participation in in-person sessions [3]. The convenience of telerehabilitation may also lead to higher attendance rates for individuals with busy schedules or other commitments. It may also be a less expensive alternative to in-person rehabilitation due to decreased travel expenses [4].

Multiple systematic reviews have shown the effectiveness of telerehabilitation; for instance, Dias et al. found high-quality evidence that telerehabilitation was not different from other interventions for adults with physical disabilities in terms of improvements in pain, physical function, and long-term quality of life [5-9]. Due to the remote nature of telerehabilitation, patients cannot receive immediate physical assistance from rehabilitation providers if they experience an adverse event. Adverse events are defined as “negative consequences of care that result in unintended injury or illness which may or may not have been preventable” [10]. For instance, they may include falls during physiotherapy or aspiration pneumonia due to speech language pathology swallowing assessments [11, 12]. There can also be safety risks related to privacy, as personal health information is being transmitted across digital platforms. There is a paucity of research surrounding the patient safety of telerehabilitation, potentially contributing to its limited uptake among clinicians and patients [139]. While many individual studies include safety data, there exists a research gap as there has yet to be any synthesis of the existing literature that summarizes the currently available research on adverse events related to telerehabilitation. There has been a prior scoping review on measures to ensure safety during telerehabilitation for patients with stroke, specifically, but the current review differs as it focuses on adverse events and encompasses all health/chronic conditions that could be served by telerehabilitation [14. This scoping review aims to conduct a systematic search of published literature on adverse events during the delivery of telerehabilitation and map out the extent of existing research. The WHO recognizes patient safety as a global health priority, and notes that investing in patient safety is important for health outcomes, cost reduction related to patient harm, and health system efficiency [15]. It is important to understand how adverse events are associated with telerehabilitation delivery, so that safety precautions and risk-mitigation measures can be thoughtfully planned and implemented, to optimize its uptake and delivery. Knowledge of the safety of telerehabilitation can help patients make more informed decisions, aid in clinical and funder decision-making and inform safety quality indicators for telerehabilitation.

Comment 2: While the introduction mentions the benefits of telerehabilitation, it could benefit from acknowledging some potential drawbacks or limitations, which would add context to the research question.

Response 2: Thank you for raising this. We agree that explaining the limitations of telerehabilitation would help bridge into the research question, and have highlighted them specifically in these sentences below:

However, questions remain about potential limitations of telerehabilitation, particularly regarding its safety compared to traditional in-person rehabilitation. Due to the remote nature of telerehabilitation, patients cannot receive immediate physical assistance from rehabilitation providers if they experience an adverse event. Adverse events are defined as “negative consequences of care that result in unintended injury or illness which may or may not have been preventable” [10]. For instance, they may include falls during physiotherapy or aspiration pneumonia due to speech language pathology swallowing assessments [11, 12]. There can also be safety risks related to privacy, as personal health information is being transmitted across digital platforms.

Comment 3: The citation of specific systematic reviews and their findings on the effectiveness of telerehabilitation is somewhat vague. It would be more informative to provide key findings or statistics from these reviews to support the assertion that telerehabilitation is effective.

Response 3: 

Thank you for raising this point. We have removed this section and have added in a specific study’s findings regarding the effectiveness of telerehabilitation: 

Multiple systematic reviews have shown the effectiveness of telerehabilitation; for instance, Dias et al. found high-quality evidence that telerehabilitation was not different from other interventions for adults with physical disabilities in terms of improvements in pain, physical function, and long-term quality of life [5-9].

Comment 4. The introduction could be more concise and focused on setting up the research question and rationale.

Response 4: Thank you for the suggestion. The introduction section has been modified as per comment/response 1 to be more concise and focused on setting up the research question and rationale. 

Methods:

Comment 1. While the methods section mentions adherence to the JBI framework and PRISMA-ScR guidelines, it lacks a detailed explanation of how these frameworks will be applied in the review. Clarification on the specific steps and criteria used would enhance the transparency of the review process.

Response 1: Thank you for pointing this out. We have included the following to better clarify how the frameworks will be applied:

This protocol will adhere to the Joanna Briggs Institute (JBI) methodological framework for scoping reviews, which provides guidance on the outline of the review, inclusion criteria (i.e., PCC), search strategy, extraction, presenting and summarizing the results, and any potential implications of the findings for research and practice [162]. The reporting of the scoping review will adhere to the Preferred Reporting Items for Systematic Reviews and Meta-Analyses Extension for Scoping Reviews (PRISMA-ScR) guidelines, to ensure all the components of a high-quality scoping review are completed and a filled checklist will be viewable in the Appendix [176]. Our team includes members with extensive experience in scoping reviews and telerehabilitation. 

Comment 2: The inclusion criteria mention studies published between 2013 and 2023, but the rationale for this time frame is not clearly explained. Providing a justification for this specific time frame would improve the clarity of the methodology.

Response 2:

Thanks for the suggestion. The rationale for the inclusion criteria has been clarified as below:

Studies will be limited to those published between 2013-2023, because a study by Zheng et al. found that 2013 was the start of a more significant development period of telerehabilitation, with only a few papers on telerehabilitation published prior [18]. Additionally, the year 2013 marked the emergence of video communication technologies such as Zoom and Google Hangout that are commonly used in telerehabilitation today, which will ensure that the review’s results are relevant to the current practice of telerehabilitation [19].

Comment 3. The search strategy is briefly mentioned but lacks the actual search terms or keywords used. Including the search terms would make the methodology more reproducible.

Response 3: Thank you for the comment. We have included the full MEDLINE ALL search strategy with search terms in the Appendix. The search terms were also described in the manuscript as seen below: 

Search terms included Medical Subject Headings (MeSH), EMTREE terms, American Psychological Association thesaurus terms, and CINAHL headings and text words to capture concepts and synonyms of telerehabilitation and adverse events. Results were limited to the English language and the publication period from 2013 to present. The full MEDLINE ALL search strategy can be viewed in S1 Appendix.

Comment 4. The discussion of data analysis and presentation is limited. It would be helpful to provide more information on how the data will be synthesized quantitatively and how content analysis will be conducted.

Response 4: 

Thank you for pointing this out. This section has been expanded as seen below:

Data from this scoping review will be summarized quantitatively using numerical counts and qualitatively via content analysis, based on best practices for reporting of scoping reviews [21]. The data will be grouped by intervention type and by type of adverse event (physical, social, psychological), and coded and analyzed manually. Numerical counts and content analysis will be used to reveal trends in the data such as the most common method of telerehabilitation, the health condition with the most adverse events, and the frequency of different types of adverse events. Synthesis will occur in duplicate and independently.

Comment 5. The mention of contacting authors for missing data raises questions about how this process will be carried out and how it might affect the review's completeness. A brief explanation of the approach to contacting authors would be beneficial.

Response 5: Thank you for drawing attention to this. We agree that this process may not be feasible, and the effort involved will have limited value. It has been removed. 

Discussion Section:

Comment 1. While the dissemination plan is mentioned, it lacks specificity regarding how the results will be translated to different audiences (healthcare professionals, researchers, policymakers, etc.). It would be beneficial to outline concrete strategies or channels for knowledge translation to ensure that the research reaches its intended audience effectively.

Response 1: Thank you for the comment. The dissemination plan has been edited as follows to include concrete strategies and channels for knowledge translation:

The dissemination plan of this review includes traditional knowledge translation approaches of journal publication and conference presentations. This scoping review will be published in a relevant peer-reviewed academic journal such as PLOS One, Annals of Physical and Rehabilitation Medicine, Journal of Telemedicine and Telecare, or Telemedicine and eHealth, as they are reputable journals that publish articles within the same field as our review. This would help reach healthcare professionals, researchers, policymakers, and other relevant knowledge users who are actively searching for papers related to telerehabilitation and patient safety. The results from this review will be presented at conferences such as the American Telemedicine Association (ATA) Annual Conference, International Society of Physical and Rehabilitation Medicine (ISPRM) World Congress, American Congress of Rehabilitation Medicine, or International Conference on Telemedicine and Telehealth. This will further help reach the intended audience, including those who are not actively searching for papers on the topic. We will also leverage social media online platforms to increase the visibility of the research article and reach a broad population who may not necessarily be up to date with academic journals or related conferences. This paper will also be presented at the Canadian Telerehabilitation Community of Practice, which is a forum for frontline clinicians. We will also engage with patient partners to elucidate the most effective ways of dissemination to this important knowledge user group.

Comment 2. The manuscript lists several potential journals and conferences for publication and presentation, but it would be more informative to explain the rationale behind the selection of these specific outlets. Why were these journals and conferences chosen, and how do they align with the scope and goals of the review?

Response 2: Please see the prior comment (Discussion section Comment 1) for the full paragraph, but the relevant edited excerpt is seen below:

This scoping review will be published in a relevant peer-reviewed academic journal such as PLOS One, Annals of Physical and Rehabilitation Medicine, Journal of Telemedicine and Telecare, or Telemedicine and eHealth, as they are reputable journals that publish articles within the same field as our review.

Comment 3. The discussion section mentions the strengths of the review, such as the use of the JBI framework, adherence to reporting guidelines, and the expertise of the review team. However, it would be helpful to expand on how these strengths contribute to the rigor and quality of the review. How will the JBI framework enhance the review process, for example?

Response 3: 

Thanks for the suggestion. The section has been expanded as seen below: 

This review has a number of strengths. For instance, all study phases (screening, extraction, synthesis) will be completed in duplicate and independently. This review is also guided by the JBI methodological framework for scoping reviews and will adhere to the PRISMA-ScR reporting guidelines [16, 17]. These frameworks and guidelines were developed by internationally recognized experts in scoping review methodology for the development of high-quality scoping reviews. Following these recommendations will ensure all the components of a rigorous scoping review are completed. Furthermore, our team includes members with extensive experience in telerehabilitation and conducting scoping reviews. 

Comment 4. While the expected limitations are briefly mentioned, the discussion could benefit from further exploration of how these limitations might impact the review's findings and implications. For instance, how might the exclusion of gray literature affect the comprehensiveness of the review?

Response 4: Thank you for pointing this out. We have edited the section as seen below: 

Expected limitations of the study design are the exclusion of gray literature and studies not published in the English language. Excluding studies not published in the English language may reduce our understanding of telerehabilitation-related adverse events in limited resource countries. The exclusion of gray literature may lead to purely clinical settings and very recent data to be excluded from the review. 

Comment 5. The discussion mentions that no risk of bias assessment will be conducted due to the scoping review's nature. While this is acceptable for a scoping review, it would be helpful to briefly explain why assessing risk of bias is not applicable in this context:

Response 5: Thank you for the comment. We have expanded on it below.

In addition, no risk of bias assessment will be conducted as the aim of a scoping review is to map out the extent of literature. This means that this review will provide an overview of the literature and identify knowledge gaps but will not assess the methodological quality or risk of bias of the individual studies, as that is not the purpose of a scoping review [17].

Comment 6. The discussion outlines the importance of understanding adverse events related to telerehabilitation, especially in the context of the COVID-19 pandemic. However, it could be strengthened by providing specific examples of how these findings might inform government funding or guide policymakers. Concrete examples would enhance the practical implications of the review.

Response 6: 

Thank you for your suggestions. We have added the following specifics to enhance the practical implications of this review:

A greater understanding of the safety and optimization of telerehabilitation can help influence government funding and guide policymakers regarding whether telerehabilitation should be offered as part of standard care, which is important as a lack of leadership and organizational support has been found to hinder implementation of telerehabilitation [22]. These findings may also inform resource allocation decisions such that safer forms of telerehabilitation can be prioritized for funding. The results may also suggest the need for telerehabilitation competency training for healthcare workers to ensure quality and safe care. The scoping review results will also inform researchers on how to optimize the safety of existing and emerging telerehabilitation technologies. From a patient perspective, findings from this review may aid in decision-making regarding participation in telerehabilitation.

Comment 7: Overall, the discussion section could be improved by providing more specific details regarding knowledge translation, rationale for journal and conference selection, and a deeper exploration of the implications of expected limitations. Additionally, offering concrete examples of how the review's findings could impact decision-making, and policy would enhance the section's clarity and impact.

Response 7: Thank you for the summary of suggested edits. We hope this new iteration has improved upon these sections.

We hope that we have satisfactorily addressed the reviewers’ comments. Please do not hesitate to contact us if you require any further information. Thank you and have a wonderful day.

Kind Regards,

Sarah Munce, PhD on behalf of the Team

Toronto Rehabilitation Institute, University Health Network

345 Rumsey Road, Toronto, Ontario, M4G 1R7, Canada

E-mail: sarah.munce@uhn.ca

---

## [Decision Letter · Decision Letter 1]

16 Jan 2024

Adverse events associated with the delivery of telerehabilitation: A scoping review protocol

PONE-D-23-22135R1

Dear Dr. Sara,

We’re pleased to inform you that your manuscript has been judged scientifically suitable for publication and will be formally accepted for publication once it meets all outstanding technical requirements.

Kind regards,

Kalyana Chakravarthy Bairapareddy, PhD

Academic Editor

PLOS ONE

Additional Editor Comments (optional):

Please verify the references as suggested by the reviewer while submitting the revised manuscript.

Reviewers' comments:

Reviewer's Responses to Questions

**Comments to the Author**

1. Does the manuscript provide a valid rationale for the proposed study, with clearly identified and justified research questions?

Reviewer #2: Yes

2. Is the protocol technically sound and planned in a manner that will lead to a meaningful outcome and allow testing the stated hypotheses?

Reviewer #2: Yes

3. Is the methodology feasible and described in sufficient detail to allow the work to be replicable?

Reviewer #2: Yes

4. Have the authors described where all data underlying the findings will be made available when the study is complete?

Reviewer #2: Yes

5. Is the manuscript presented in an intelligible fashion and written in standard English?

Reviewer #2: Yes

6. Review Comments to the Author

You may also provide optional suggestions and comments to authors that they might find helpful in planning their study.

Reviewer #2: Revised manuscript stands good with satisfactory responses from the author. Please check citation for the followings sentences:

There is a paucity of research surrounding the patient safety of telerehabilitation, potentially contributing to its limited uptake among clinicians and patients [139]

This protocol will adhere to the Joanna Briggs Institute (JBI) methodological framework for scoping reviews, which provides guidance on the outline of the review, inclusion criteria (i.e., PCC), search strategy, extraction, presenting and summarizing the results, and any potential implications of the findings for research and practice [162].

Appendix [176]

7. PLOS authors have the option to publish the peer review history of their article (what does this mean?). If published, this will include your full peer review and any attached files.

Reviewer #2: No

---

## [Editor Report · Acceptance letter]

12 Feb 2024

PONE-D-23-22135R1 

PLOS ONE

Dear Dr. Munce, 

I'm pleased to inform you that your manuscript has been deemed suitable for publication in PLOS ONE. Congratulations! Your manuscript is now being handed over to our production team.

Kind regards, 

on behalf of

Dr. Kalyana Chakravarthy Bairapareddy 

Academic Editor

PLOS ONE